# Recent Progress in Diatom Biosilica: A Natural Nanoporous Silica Material as Sustained Release Carrier

**DOI:** 10.3390/pharmaceutics15102434

**Published:** 2023-10-09

**Authors:** Hayeon Lim, Yoseph Seo, Daeryul Kwon, Sunggu Kang, Jiyun Yu, Hyunjun Park, Sang Deuk Lee, Taek Lee

**Affiliations:** 1Department of Chemical Engineering, Kwangwoon University, 20 Kwangwoon-ro, Nowon-gu, Seoul 01897, Republic of Korea; noeydla@kw.ac.kr (H.L.); akdldytpq12@kw.ac.kr (Y.S.); rtr2001@kw.ac.kr (S.K.); yjun2473@kw.ac.kr (J.Y.); andy9760@kw.ac.kr (H.P.); 2Protist Research Team, Microbial Research Department, Nakdonggang National Institute of Biological Resources (NNIBR), 137, Donam 2-gil, Sangju-si 37242, Republic of Korea; kdyrevive@nnibr.re.kr

**Keywords:** diatom, drug delivery systems, biosilica, oral administration, biocompatible

## Abstract

A drug delivery system (DDS) is a useful technology that efficiently delivers a target drug to a patient’s specific diseased tissue with minimal side effects. DDS is a convergence of several areas of study, comprising pharmacy, medicine, biotechnology, and chemistry fields. In the traditional pharmacological concept, developing drugs for disease treatment has been the primary research field of pharmacology. The significance of DDS in delivering drugs with optimal formulation to target areas to increase bioavailability and minimize side effects has been recently highlighted. In addition, since the burst release found in various DDS platforms can reduce drug delivery efficiency due to unpredictable drug loss, many recent DDS studies have focused on developing carriers with a sustained release. Among various drug carriers, mesoporous silica DDS (MS-DDS) is applied to various drug administration routes, based on its sustained releases, nanosized porous structures, and excellent solubility for poorly soluble drugs. However, the synthesized MS-DDS has caused complications such as toxicity in the body, long-term accumulation, and poor excretion ability owing to acid treatment-centered manufacturing methods. Therefore, biosilica obtained from diatoms, as a natural MS-DDS, has recently emerged as an alternative to synthesized MS-DDS. This natural silica carrier is an optimal DDS platform because culturing diatoms is easy, and the silica can be separated from diatoms using a simple treatment. In this review, we discuss the manufacturing methods and applications to various disease models based on the advantages of biosilica.

## 1. Introduction

In chemotherapy, exploring new drugs and reducing their side effects by improving their bioavailability based on various formulations, such as pills and creams, are essential [1,2]. Many drugs may randomly damage organs other than those targeted during administration, owing to poor solubility and lack of target function, a complication requiring alleviation in various pharmaceutical fields [3,4,5]. Therefore, drug delivery system (DDS) studies investigate effectively delivering drugs with various characteristics beyond the physiological barrier in the body through various delivery vehicles [4,5].

DDSs have been continuously developed since the introduction of the concept of sustained release systems in the 1950s, and numerous DDS platforms have been developed to overcome the physiological barriers in various administration routes [6]. As a representative example, DDS platforms, such as collagen-based oral delivery pills, are resistant to digestive enzymes, and patches are used for sustained drug delivery through the skin [7]. Additionally, nano- and microscale DDS platforms for highly functional delivery systems have been developed in recent decades. A decrease in the DSS size improves various functions, such as solubility of poorly soluble drugs, evasion of the body’s immune system, cell permeability, and passive cancer targeting based on enhanced permeability and retention effects [8,9].

Sub-microscale DDS platforms are currently developed primarily through chemical synthesis methods based on inorganic materials, such as silica and metals, or various organic materials, such as polymers and lipids [10]. Among them, the mesoporous silica-based DDS (MS-DDS), which primarily has a nanosized porous structure on its surface, has been recognized as safe by the US Food and Drug Administration and has been evaluated as an effective DDS platform for various drug deliveries [11]. MS-DDS is currently evaluated as an excellent oral DDS owing to its porous structure-based controlled release ability and high drug solubility for poorly soluble drugs [12]. Among MS-DDS platforms, artificially synthesized mesoporous silica nanoparticles (MSN) have the advantage of nanosized mesoporous carriers; however, they have commercialization limitations, such as cytotoxicity and scale-up, owing to various toxic substances used during their production (Figure 1) [13]. Hence, researchers are seeking novel MS-DDS platforms that overcome the disadvantages of synthetic MS-DDS.

The frustule of diatoms in various water systems has been an attractive natural MS-DDS, similar to biosilica. Diatoms, the microalgae in various aquatic environments, are single-celled autotrophic protists, with a size of 1–100 μm, surrounded by a double silica wall. They contain chlorophyll and can produce nutrients via photosynthesis [14,15]. In addition, diatoms are believed to contribute up to 25% of global primary productivity, equivalent to that of tropical rain forests [16,17]. Furthermore, they are essential primary producers in aquatic systems, accounting for approximately 40% of marine primary productivity, and are reportedly essential in the ocean silicon cycle [18,19]. Diatom cell walls comprise silica, which has a complex and unique structure depending on the species and is used to identify the species in morphological classification. To date, approximately 18,000 and 2260 species have been reported worldwide [20] and in Korea [21], respectively.

Each biosilica diatom varies in size and shape (e.g., disk-, rod-, and linear-shaped) depending on the species, and the MSN-like porous nanostructures are formed by unique enzymes involved in silica fixation and synthesis [13,22]. These features enable controlled drug release even during long-term digestion, facilitating their use in developing oral formulations [23,24]. Moreover, most carriers developed to date are formed through artificial synthesis; however, this biosilica is reported to be naturally formed during the growth of organisms and has excellent biodegradability in the body [25,26]. In addition, diatoms can be easily cultured and purified to obtain biosilica of uniform size, and according to recent MS-DDS studies, these natural silica carriers can be considered excellent oral DDS platforms [23,27]. In this review, the latest study trends and the performance of biosilica in various drug administration methods are analyzed. Similarly, we discuss the potential of biosilica for developing an effective oral DDS platform.

**Figure 1 pharmaceutics-15-02434-f001:**
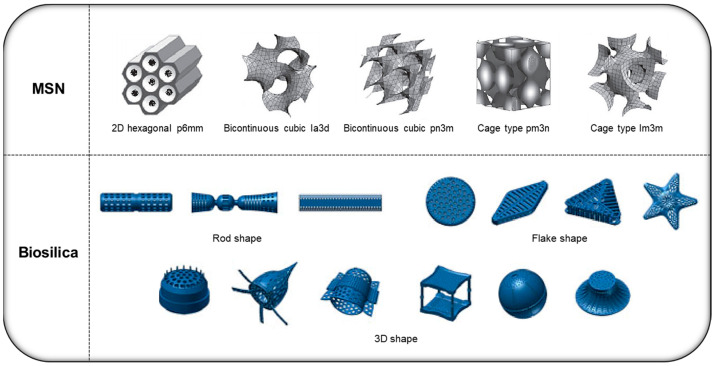
Various structures of the two types (synthetic/natural) of mesoporous silica-based drug delivery system (MS-DDS). Reproduced with permission from [28], published by the National Center for Biotechnology Information, 2015, and [29], published by Springer, 2012.

## 2. Various Types of Silica-Based DDS Platforms

An MS-DDS is an excellent DDS used in various medical fields because of its high specific surface area, uniform nanosized pores, improved drug loading capacity, and ability to solubilize slow-release and poorly soluble drugs [30,31,32]. The primary sources of MS-DDSs are MSN and diatom biosilica, and the synthesis methods and characteristics of both MS-DDS platforms are described here.

### 2.1. MSN

MSN is primarily synthesized using surfactants, and the general synthesis method involves a polymer template (Figure 2). This method is conducted using a sol–gel method, where a silicon alkoxide precursor is hydrolyzed and condensed in the presence of an acid or base catalyst, using a self-assembled polymer as a template. To remove the template after transferring the polymer structure to an inorganic material, a porous structure is formed using a surfactant as the structure inducing the agent. Polycondensation proceeds and the structure changes to a sol, gel, or colloidal solution, depending on the conditions, and spherical silica particles are produced under dilute conditions [33,34]. In addition, MSN is synthesized using the evaporation-induced self-assembly method, which involves changing the concentrations of the precursor and surfactant via alcohol evaporation. Template analogs of the silica precursors are synthesized with varying concentrations, and surfactants and silica surfactants are assembled when ethanol evaporates from an ethanol–water solvent. Similarly, there is a simple quenching approach for synthesizing MSN that involves varying the pH using excess water and dilute hydrochloric acid [35].

Furthermore, selective etching strategies can be used to synthesize dense MSNs on pure silica frameworks and other organic–inorganic composites, producing a three-layer MSN. The core and outermost shell layers of the particle have a hydrolyzed pure silica structure, and the middle layer has a hybrid silica structure where organic and inorganic materials are combined. All three layers are treated with tetraethyl orthosilicate [36].

An MSN synthesized in this manner can be used for several diseases because of their unique properties, such as large surface area and pore volume, controllable particle size, good biocompatibility, and facile functionalization chemistry [37,38]. However, using toxic substances, such as surfactants (e.g., cetyl trimethyl or ammonium bromide), MeOH, or expensive silica materials (MCM-41 or SBA-15), in synthesizing MSN hinders commercialization [39,40].

### 2.2. Diatom Biosilica

Biosilica (in the frustule of diatoms) is formed based on silicification through silica polymerases involved in the processes of fixation, transport, and stabilization of silicic acid, with the growth of diatoms. It has a unique nanoporous structure aligned in three dimensions by enzymes forming silica matrix structures, such as transmembrane proteins or silaffins in silica deposition vesicles [41,42,43,44]. Similar to MSN, this nanostructure significantly affects its high drug loading capacity and sustained release [45,46,47]. In addition, this natural silica carrier can be easily obtained through diatom culturing. Furthermore, since it can be purified through a simple method using H_2_O_2_ and HCl, it can be considered an alternative to synthetic silica (particularly MSN) in the pharmaceutical field [48]. Although biosilica has a relatively large size (in microscale) compared with MSN, it shares most of MSN’s advantages as a DDS and is currently considered a good sustained-release drug carrier in several DDS studies, owing to its highly organized hierarchical structure [22,44,49]. Table 1 showed the comparison table between diatom biosilica and MSN.

**Figure 2 pharmaceutics-15-02434-f002:**
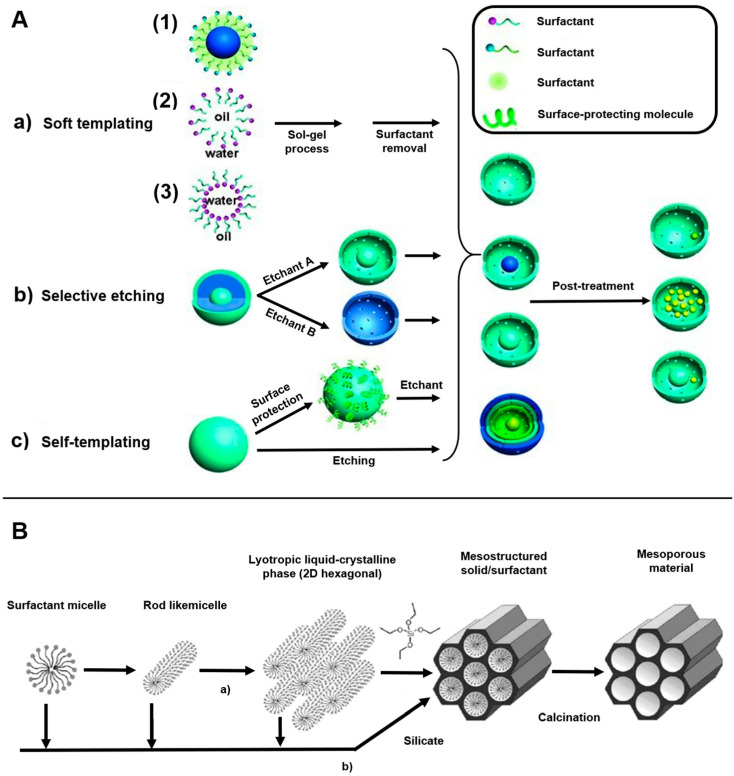
Overview of the methods for synthesizing MS-DDS for drug delivery: (**A**) (a) soft templating method, (b) selective etching strategy, and (c) self-templating method; (**B**) synthesis steps of SBA-15, an example of mesoporous nanosilica (LCTA technology). Reproduced with permission from [36], published by Wiley Online Library, 2012, and [50], published by Elsevier, 2017.

### 2.3. Frustule of Diatoms

The traditional classification of diatoms is based exclusively on the morphological characteristics of the frustule [79]. The frustule comprises two valves: the epivalve and the hypovalve (Figure 3). The epivalve is slightly larger than the hypovalve. They are joined by girdles or cingulae (a series of silica bands), forming a Petri dish-like structure [80]. The wall of the frustule is perforated by areolae (pores), whereas a thin, porous silica layer, called a velum, occludes the inner side of the wall. Centric diatoms have a radiating pore arrangement, whereas, in pennate diatoms, pores are arranged in an elongated and bilaterally symmetric pattern [81]. Contrasting pore areas, such as ocelli and apical pore fields in centric and pennate diatoms, respectively, are associated with attachment and/or colony development by secreting mucilage [82].

## 3. Application of MS-DDS Platforms According to Various Administration Routes

Depending on the disease model and drug characteristics, MS-DDS platforms are delivered to the target site through various routes, such as oral, transdermal, and injection. MS-DDS is attractive, particularly for oral use, owing to its porous structure and sustained release [23,24,83]. Furthermore, because the surface of MS-DDS, comprising SiO_2_, can be easily modified, high functionality can be imparted to the carrier through various chemical treatments [84,85]. The aforementioned characteristics of the MS-DDS suggest that it can be applied to various medical fields. In this section, we examine the current situation, focusing on biosilica.

### 3.1. Oral

Oral administration is a widely used drug administration route owing to advantages such as patient convenience, large-scale manufacturing, and cost-effectiveness [23,86]. However, drugs used for oral administration pass through the gastrointestinal tract (GIT) and are affected by the low stomach pH, various digestive enzymes, and detoxifying enzymes, such as cytochrome P450 [87,88]. Therefore, a DDS for oral administration should safely protect the drug from these in vivo interfering factors [83,89]. In addition, the carrier should have low cytotoxicity and be biodegradable to enable its excretion [23,25]. Several DDS studies have been conducted to address these oral administration challenges, and various drug delivery studies have been performed based on the excellent sustained release and low cytotoxicity of biosilica among MS-DDS platforms [13,23].

Aw et al. (2012) used diatomaceous earth (DE) from sedimentary rocks for implantation and oral drug delivery [13]. Through various spectroscopic techniques (energy-dispersive X-ray spectroscopy, X-ray powder diffraction spectroscopy, and thermogravimetry) and imaging techniques (scanning electron microscopy; SEM), it was confirmed that the drug was physically adsorbed to the surface, pores, and the internal hollow structure of diatom biosilica. This team modified the surface of the diatom with organosilanes to control its hydrophilicity and hydrophobicity, producing a loading capacity of approximately 21.99 ± 2% and an encapsulation efficiency of 94.0 ± 1%. In addition, 70% of the drug that precipitated on the surface was rapidly released during burst release for 6 h, and continuous release was observed through sustained release for 2 weeks. This sustained release was presumed to be owed to the drug loading in the hollow structure inside the biosilica (Figure 4A).

López-Cebral et al. (2018) studied the sublingual administration of drugs using β-chitosan and DE. They used a β-chitosan–DE composite membrane, based on the electrostatic interaction between the positive charge of β-chitosan and the negative charge of the mucus layer, for drug delivery. Hydrophilic (gentamicin) and hydrophobic (dexamethasone) drugs were loaded onto the fabricated composite membrane, and the drug was delivered sublingually. Concerning the hydrophilic drug, an immediate reaction of the drug was achieved, with 100% loading and release within the first 10 min, whereas the hydrophobic drug had a relatively low loading rate of 79.6 ± 8.1% and a release of approximately 40% after 8 h, confirming the differences in hydrophobic and hydrophilic drug delivery capabilities [84].

Zhang et al. (2013) used mesalamine and prednisone, which are commonly used to treat gastrointestinal (GI) disorders, for oral drug delivery (Figure 4B) [23]. In addition, a cell viability test based on colon cancer cells (Caco-2, HT-29, and HCT-116) confirmed that biosilica had little cytotoxicity below 1 mg/mL (Figure 4C) [23,48]. It was assumed that the low toxicity of the biosilica was influenced by the size of the diatoms (length 10–20 μm, diameter 10 μm) and the cylindrical shape of the species, which reduced its absorption rate into cells. Similarly, the strong negative charge of drug-biosilica reportedly caused minimal damage to cell membranes with the same charge and prevented particle aggregation. Under simulated GI conditions, drug-biosilica showed an excellent sustained release, and drug permeation across the Caco-2/HT-29 monolayer-based drug permeation experiments demonstrated that biosilica improved the cell barrier permeability of the drugs.

In addition, liposomes were coated onto the MS-DDS surface in a study that attempted to lower the cytotoxicity of an MS-DDS. Mudakavi et al. (2014) used liposome-coated MSN (L-MSNs) with lipids as an oral delivery system for ciprofloxacin (Cip), an antibiotic for removing Salmonella. The delivery efficiency was increased by loading Cip onto MSN, and the cytotoxicity of the delivery system was reduced by coating it with liposomes via sonication. The excellent antibacterial efficacy of Cip L-MSNs was demonstrated through in vivo experiments using mice infected with Salmonella. The delivery of Cip using Cip L-MSNs showed antibacterial efficacy even at lower concentrations than when used alone. Similarly, a macrophage-based MTT assay confirmed that L-MSNs had a higher cell viability than MSN, proving that liposome coating alleviates MSN cytotoxicity [27].

### 3.2. Rectal

Rectal drug delivery is a useful alternative primarily used for delivering drugs to the rectum in patients with a non-progressive disease, for whom oral drug delivery is not feasible [90,91]. It is used for topical treatments, such as laxatives, antipyretics, and hemorrhoids. Studies on the direct delivery of drugs to systemic circulation have been recently conducted. The rectal route can protect enzymatically labile drugs, readily absorb small molecules, and minimize first-pass metabolism when suppositories are administered at appropriate distances from the rectum [92]. In addition, in delivering stimulants, such as non-steroidal anti-inflammatory drugs, it prevents exposure of the gastric mucosa to the drug [93].

Delasoie et al. (2018) used DE as a transporter for the anticancer drug tris-tetraethyl [2,2′-bipyridine]-4,4′-diamine-ruthenium (II) complex for treating colorectal cancer. In this study, by focusing on the characteristics of cancer cells, which require more vitamins than normal cells, the biosilica surface was modified with vitamin B12 to improve its ability to target cancer cells. SEM images confirmed that the cancer cell attachment rate of the anticancer diatom carrier coated with B12 was at least thrice higher than that of the unmodified diatoms. In addition, tests were performed using simulated gastric fluid (SGF) and colon fluid (SCF) to evaluate the resistance to vitamin B12 coating. After immersing DEMs-B12-1 in SGF and SCF for 2 h, approximately 56% and 18% of the coatings were degraded, respectively. The SGF experiment confirmed that the vitamin coating of the transporter was maintained at ≥45% under human GI conditions. Similarly, the higher resistance in the SCF experiment suggests that the sea otter delivery system could be used both as a suppository and for oral administration [94]. However, in rectal drug delivery, drug absorption is irregular, and the surface area available for absorption is limited. Moreover, there is a dissolution challenge owing to the low body fluid content in the rectum, and the most critical disadvantage is that patient compliance is low [95].

### 3.3. Skin

Since the 1970s, the skin has been a utilized route for topical and systemic medications, with the advent of transdermal patches [96]. The skin is the outermost organ of the body and provides a natural physical barrier against the penetration of foreign substances. Skin administration eliminates many side effects of parenteral administration [97,98]. It enables continuous drug delivery into the blood circulation and provides a comfortable alternative route for drug administration [99].

Nafisi et al. (2018) used functionalized MSN (MCM41) as a drug carrier for transdermal delivery of the local anesthetic lidocaine (Lido). Lido is a local anesthetic commonly used as an injection [62]. They proposed using Lido/MCM41-NH2, surface-modified with positively charged aminopropyl groups, to enhance drug delivery efficiency based on the electrostatic mutual attraction of MCM41. As a result of zeta potential measurement, Lido/MCM41 has a charge of −19 mV, and Lido/MCM41-NH2 has a charge of 32 mV. These results confirmed that the surface of Lido/MCV41 was positively charged by -NH2. Cell and in vitro experiments on human skin have shown that Lido/MCM41-NH2 is more permeable to the skin than Lido/MCM41 and free Lido, because of the electrostatic interactions between the negatively charged skin membrane cells (Figure 5A) [100]. This suggests that an MSN surface modified with -NH2, can increase the delivery efficiency of Lido through the skin.

Sapino et al. (2017) suggested using a nanoparticle similar to MCM 41 for drug delivery via skin administration. Silica nanoparticles have been used to topically administer methotrexate (MTX), an immunosuppressant drug for epidermal cells, to reduce the side effects of another drug (Figure 5B) [65,101]. MSNs, loaded with MTX (MTX/MSN), were prepared using the impregnation method. To demonstrate the drug delivery efficiency of the MTX/MSNs, HaCaT cells (keratinocytes) were treated with MTX and MTX/MSNs. MTX and MTX-MSN inhibited cell growth by over 70%, and MTX-MSN effectively delivered MTX to the cells. Franz cell experiments on porcine skin showed that MSN enhanced the ability of MTX to penetrate the stratum corneum. Moreover, shea butter was added to the carrier to enhance drug penetration.

Lee et al. (2020) studied the hemostatic effects of synthetic silica and biosilica powders. Biosilica was extracted from *Melosira nummuloides*, collected from the lava seawater on Jeju Island. In this study, MSN had a poor hemostatic effect because blood could not penetrate it owing to its hydrophobic surface, whereas biosilica had superhydrophilic and superhemophilic properties, showing effective hemostatic reactions in vitro and in vivo (Figure 5C) [102]. The biosilica powder had a chemical composition and size similar to those of MSN. However, biosilica powder containing abundant silanol (SiOH) immediately absorbed blood, became a viscoelastic solid when wet, and enhanced the activation of coagulation factor XII, showing superior hemostatic effects to synthetic silica. Similarly, they reported that the superhydrophilicity of biosilica was not created by the nanoporosity of the diatoms but by the synergistic effect of the high-density silanol anions and nanostructures. In addition, Feng et al. (2016) improved the hemostatic properties of biosilica by coating its surface with chitosan to increase its absorption. Furthermore, higher concentrations of polar silanol groups created more negatively charged interfaces, effectively promoting blood coagulation [103].

Rozan et al. (2022) loaded the antibiotic doxycycline (DOXY) into biosilica to treat integumentary wounds [104]. They confirmed that biosilica/hydrogels have a slower decomposition rate than general hydrogels and that biosilica contributes to maintaining their shape by delaying the decomposition rate of hydrogels. Based on these biosilica characteristics, a hybrid DDS called DOXY/biosilica/hydroxybutyl chitosan (HBC), a hydrogel for wound protection and healing, was produced. In an in vitro antibacterial activity test, the DDS showed 100% and 98% inhibition of *Staphylococcus aureus* and *Escherichia coli*, respectively. In addition, in an in vivo experiment conducted on mouse skin, 99.4 ± 0.4% of wounds caused by the hybrid DDS were almost completely closed, while approximately 11% were not closed by control free HBC and free DOXY (Figure 5D). Thus, the hybrid DDS exhibited an excellent healing effect compared with the control group. The skin has the advantage of a long application time owing to its relatively low enzymatic degradation and transdermal delivery; however, drug delivery is challenging because the biological barrier function of the stratum corneum is high [105]. Moreover, as reported in previous studies, skin permeability is lower than that of the intestinal epithelial cell membrane [106].

**Figure 5 pharmaceutics-15-02434-f005:**
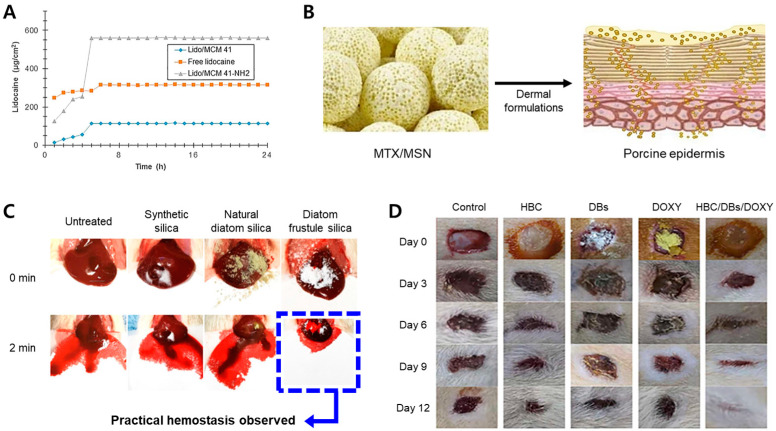
Cases of applying the mesoporous silica drug delivery system (MS-DDS) platform to skin. (**A**) Ex vivo skin permeation profile of free lidocaine and lidocaine loaded mesoporous silica nanoparticles (MSN) complexes; (**B**) methotrexate (MTX)/MSN application to porcine epidermis; (**C**) in vivo hemostatic ability of diatom frustule silica; (**D**) wound healing using HBC, DB, DOXY, and HBC/DB/DOXY hydrogels on days 0, 3, 6, 9, and 12. Reproduced with permission from [62,65,104], published by Elsevier, 2017, 2018, 2022, and [102], published by the American Chemical Society, 2020.

### 3.4. Injections

Injection types include intravenous, intramuscular, and the most commonly used, subcutaneous injections, and some drugs should be given only through them [107]. Among them, the subcutaneous route is used for administering various drugs because of its high bioavailability and rapid action expression.

Delalat et al. (2015) developed functional biosilica that can target cancer cells exposed to the antibody-binding domain (IgG) in the biosilica of the diatom *Thalassiosira pseudonana* through genetic modification. To improve the drug delivery efficiency of SN38, an anticancer drug, the drug was contained in cationic liposomes based on cationic lipids and bound to the negatively charged biosilica surface through electrostatic interactions. By delivering anticancer drugs to cancer cells through genetically modified biosilica, neuroblastoma and B-lymphoma cells were selectively killed [108].

Wu et al. (2017) synthesized MSN-based metal–organic frameworks (mesoMOFs) and used them as carriers for the anticancer drug doxorubicin (DOX) [109]. Metal–organic frameworks (MOFs) are porous materials with a high specific surface area, enabling inorganic metal ions or clusters to interconnect by organic ligands. MOFs are attractive as novel drug delivery systems [110,111]. The MTT assay confirmed that empty mesoMOFs were nontoxic to 4T1 breast cancer cells and 3T3 fibroblasts and that DOX-loaded mesoMOFs showed a higher apoptosis rate in 4T1 breast cancer cells than free DOX. In an in vivo test, breast cancer-bearing mice were divided into groups administered saline (control), blank mesoMOFs, DOX–HCl, and DOX-loaded mesoMOFs and observed for 27 days. Among the four groups, the tumor suppression effect was the greatest in the DOX–HCl- and DOX-loaded mesoMOF groups, and the suppression effects in the two groups were similar. However, in the DOX–HC1 group, the suppression effect decreased on the 12th day, and systemic toxic side effects were observed. The body weights changed slightly in the other three groups, including the DOX-loaded mesoMOFs. This indicates that the mesoMOF delivery carriers can mitigate the side effects of the systemic toxicity of DOX.

## 4. Value of Diatom-Based DDS Platforms

Many DDS studies using biosilica have suggested that this natural silica carrier can efficiently deliver drugs with various characteristics via various administration routes in the human body. In addition, based on the sustained and controlled release of drugs, the most essential feature of an MS-DDS, biosilica is currently considered a vital carrier in oral administration studies. However, despite the many advantages of these natural carriers, the micro size challenges of biosilica limit direct injection [112]. For example, when biosilica is directly and repeatedly administered to an area with restricted blood flow owing to thin blood vessels, such as the vitreous cavity of the eye, silica may accumulate [61]. Therefore, as with other DDS platforms, determining and applying an appropriate administration route for biosilica is essential.

### 4.1. Limitations of Diatom-Based DDS Platforms by Injection

Injectables are the most commonly used administration method, despite the possibility of being painful when administered [107], the increased risk of infection when administered intravenously, and the affected safety because the biodistribution varies depending on the particle size and surface characteristics [112,113]. In some studies, silica particles have been observed in the liver and kidneys when biosilica was administered intravenously [112,113,114]. Borak et al. (2012) studied the elimination and biodistribution of SiO_2_ from silica particles in rats via intravenous injection. They showed that the injected biosilica was excreted through the renal excretion route; however, even after excretion, large amounts of silica particles were found in the liver, lungs, and kidney glomeruli [113]. Additionally, the minimum diameter of capillaries in the body is 4 μm, and a significant portion of silica particles of size ≥300 nm are accumulated in the lungs and liver [114]. Furthermore, in vivo biodistribution studies by Delalat et al. (2015), as a result of a single intraperitoneal injection in a mice-based in vivo test, showed no decomposed biosilica particles in the lungs; however, small amounts of particles were observed in the liver and kidneys. According to these studies, silica accumulation in the liver is presumed to be owing to particle uptake by the macrophages of the reticuloendothelial system [108].

However, recent DDS studies have reported the biodegradability and low cytotoxicity of MS-DDSs including biosilica. In animal studies using mice and rats, no noticeable inflammatory response was found in the major organs (i.e., brain, heart, kidneys, liver, and lungs) of mice injected with biosilica, and it was found that biosilica can be decomposed in the body and excreted through the kidneys [108,113,115]. Additionally, Zhai et al. (2012) confirmed that MSN injected into the body was up-taken by human umbilical vein endothelial cells and degraded in the cytoplasm and lysosomes [116]. Terracciano et al. (2019) conducted studies on the safety of DE administration with *Hydra Vulgaris*, a model organism for tissue regeneration [117]. As a result of observation using DAPI dye-based confocal microscopy imaging, cell death of *H. vulgaris* due to internalized DE was not observed [117]. Kim et al. (2014) confirmed that two different sizes of silica nanoparticles (20 nm and 100 nm) had no harmful effects on tissues and organs when repeatedly orally administered to a Sprague Dawley rat model at 500 to 2000 mg/kg for 90 days [118]. The US Food and Drug Administration (FDA) and the European Food Safety Authority (EFSA) report that amorphous silica (including biosilica) and silicates are safe at oral doses of up to 1500 mg per day [119]. Likewise, studies verifying the biocompatibility of MS-DDSs are continuously being conducted.

Additionally, MS-DDSs have unique advantages compared to other biodegradable DDS platforms. For example, lipid nanoparticles (LNPs), one of the representative biodegradable drug delivery platforms, have many advantages such as excellent biocompatibility and low cytotoxicity, but they have the disadvantage of an unintended reduction in drug delivery efficiency and potential side effects, due to the initial burst release [3,120,121]. While biosilica has a lower biodegradability than LNPs, it can be applied as a various drug carrier for various drugs through its excellent sustained release [23,30,32,44]. Moreover, it has been reported that some species have pores larger than 100 nm, so biosilica can be used as secondary carriers loading nanosized DDSs, such as LNP. Delalat et al. (2015) used a complex, made by combining LNPs with a positive surface charge through cationic lipids and biosilica with a strong negative charge, as a carrier capable of loading more drugs [13,68,108].

### 4.2. Benefits of Biosilica in Oral Administration

Biosilica is currently considered an effective DDS for oral administration, based on its excellent controlled release and high solubility of poorly soluble drugs [23]. In addition, unlike the injections mentioned in Section 4.1, regarding oral administration, biosilica accumulation in the organs is less affected [23]. Similarly, as reported in previous studies, the sustained release capability and the pH resistance of biosilica are essential elements of DDS platforms in oral formulations [122,123,124]. However, when administered orally, since the transporter passes through the GIT, protection from factors related to digesting biosilica in the body and passage through biological barriers, such as the intestinal epithelial cell layer, is essential [125,126,127]. Therefore, studies on the surface modification of biosilica have recently been conducted through various methods for improving drug loading and release characteristics, intracellular absorption, and biocompatibility of biosilica [27,55,94,128,129].

### 4.3. Surface Modification of Biosilica

SiOH groups are abundant on the biosilica surface and are easily converted into functional groups such as -NH2, -COOH, -CHO, and -SH. Therefore, the biosilica surface can be easily modified with biological or chemical moieties of various substances, such as proteins, drugs, antibodies, DNA, and aptamers [61,83].

Bariana et al. (2013) provided hydrophilic and hydrophobic DE surfaces by surface modification with 3-aminopropyltriethoxysilane, N-(3-(trimethoxysilyl)propyl) ethylenediamine, and phosphonic acids (2-carboxyethyl-phosphonic acid and 16-phosphonohexadecanoic acid) [85]. In the DE with a hydrophilic surface, a poorly soluble drug (indomethacin) tended to be released sustainably, whereas in that with a hydrophobic surface, a water-soluble drug (gentamicin) tended to be released over a long time [24,85]. In addition, Terracciano et al. (2015) improved the cellular uptake and biocompatibility of DE nanoparticles modified with 3-aminopropyltriethoxysilane through surface modification with polyethylene glycol and cell-penetrating peptides [130]. Moreover, many studies have been conducted on the physicochemical surface modification of various biosilica, such as combining diatoms with a solid self-emulsifying phospholipid suspension (SSEPS) and improving the solubility of carbamazepine, an anticonvulsant, using SSEPS as a dispersion medium [83]. Feng et al. (2016) produced an excellent hemostatic agent by coating polymeric chitosan onto the biosilica surface, using electrostatic interactions [103]. Biosilica is currently developed as an efficient DDS for various in vivo pathways. Similarly, previous biosilica-based DDS studies have primarily focused on increasing the solubility of hydrophobic drugs; however, many recent studies on biosilica have demonstrated that this natural silica carrier can load hydrophilic and hydrophobic drugs by surface modification [84,85].

## 5. Conclusions

Numerous DDS platforms have been developed to overcome the limitations of intractable disease therapies. Many DDS studies targeting various disease models have been actively constructed based on MS-DDSs with controlled release (particularly sustained-release capabilities) and high drug-loading capacities. The commercialization challenges of artificial MS-DDS (MSN), such as potential cytotoxicity and bulk-up concerns, associated with developing MS-DDS formulations, have prompted the proposition of natural diatom biosilica as a next-generation MS-DDS platform.

The diatom biosilica is naturally synthesized from diatom; therefore, it is made through a simple and environmentally friendly manufacturing process, and few other chemicals other than H_2_O_2_ are used in the purification process. In addition, cytotoxicity tests showed that this natural silica carrier exhibited low cytotoxicity and high biocompatibility. Furthermore, the excellent sustained release of biosilica, due to its unique nanoporous structure, suggests that it can be applied in various drug administration methods. Recent studies have suggested that the oral administration of MS-DDS has lower cytotoxicity than other administration routes, and biosilica is considered a vital DDS platform for developing oral formulations for poorly soluble drugs.

Although biosilica has many advantages as a sustained release DDS platform, several problems remain to be solved for commercialization. The first concerns biodegradability; while nanosized MSN has been reported to biodegrade in human umbilical vein endothelial cells, there are still few studies that have clearly analyzed how micro-scale biosilica biodegrades in the body. Studies conducted to date on this have reported that orally administered biosilica is eliminated only through excretion. Second is a lack of clinical research based on biosilica. Through a mouse-based in vivo study, it has been reported that intraperitoneal injection of biosilica does not cause acute tissue toxicity, but there are still few in vivo tests based on animal models. Therefore, in order to commercialize biosilica, sufficient animal studies must be conducted to verify its immunogenicity. The last is the optimization of biosilica production suitable for DDSs. Since the size and structure of biosilica and the growth rate differs between each diatom species, it is necessary to select species suitable for being drug carriers and optimize culture conditions for these diatoms. If these problems can be overcome with sufficient studies, this natural silica carrier will become significant in pharmaceutical studies.

## Figures and Tables

**Figure 3 pharmaceutics-15-02434-f003:**
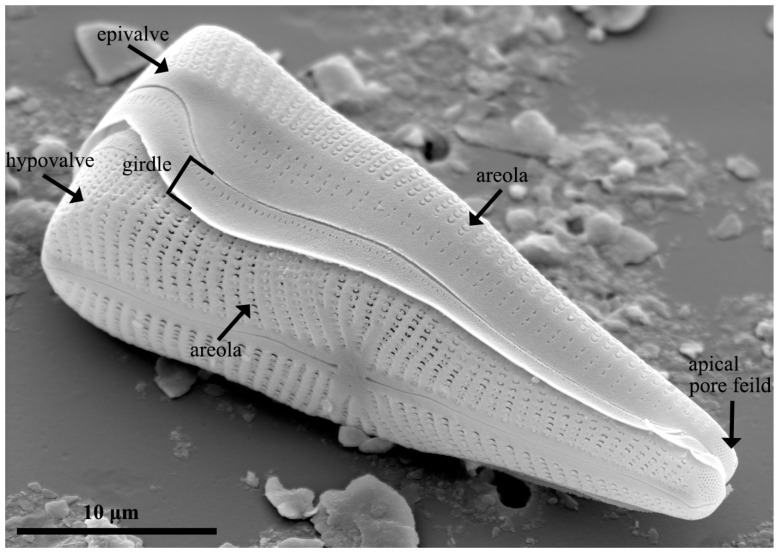
Morphology of the frustule in the diatom species *Gomphonema truncatum*.

**Figure 4 pharmaceutics-15-02434-f004:**
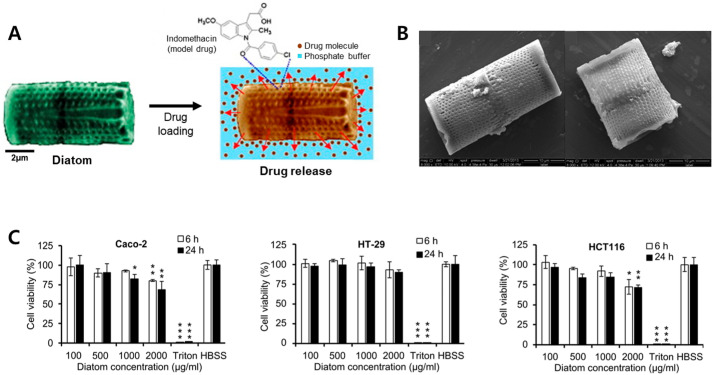
Examples of oral administration using mesoporous silica drug delivery system (MS-DDS) platform. (**A**) Indomethacin release from diatom silica microshell; (**B**) SEM images of mesalamine-loaded diatoms and prednisone-loaded diatoms. In the scale bars are 10 µm; (**C**) cytotoxicity test of diatom silica microparticles (DSMs) using a CellTiter-Glo^®^ luminescent assay with various cell models (* *p* < 0.05, ** *p* < 0.01, and *** *p* < 0.001). Reproduced with permission from [13,23], published by Elsevier, 2012, 2013.

**Table 1 pharmaceutics-15-02434-t001:** Comparison between synthetic and natural MS-DDSs.

	Diatom Biosilica	Mesoporous Silica Nanoparticle	Reference
**Surface functionalization**	Silanization	Co-condensation (direct or one-pot synthesis)/postsynthetic treatment (often referred to as grafting)	[51,52,53,54,55]
**Production**	Diatom culture, H_2_O_2_-based purification	Sol-gel, microwave synthesis, hydrothermal synthesis, template synthesis, modified aerogel methods, soft and hard templating methods, fast self-assembly, etc.	[13,56,57,58,59,60]
**Expense**	Cheap (≈$200 per tonne (DE))	Expensive	[22,44,48,49,51,61]
**Environment**	Nature (environmentally friendly, “green” material)	Toxicity (CTAB/MeOH), highly toxic and highly polluted waste is generated.	[13,44,49,60,62,63,64]
**Bulk up**	Easy	Difficult	[48,49,59,65,66,67]
**Pore size**	Alignment (diameter 30 nm to 500 nm)	Alignment (2–130 nm, typically ca. 2–6.5 nm)	[13,62,63,68,69,70]
**Particle size**	Varied (less than 350 nm on average (100 nm to 2 mm))	Varied (20 nm to 1 μm when adjusting parameters)	[36,61,69,71,72]
**Specific surface area** **Brunauer–Emmett–Teller method (BET SSA)**	Large (6–200 m^2^/g)	Large (~1000 m^2^/g)	[35,54,61,70,71,73,74,75,76,77,78]

## Data Availability

Data sharing is not applicable to this article.

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
