# Peer review of "Recent Progress in Diatom Biosilica: A Natural Nanoporous Silica Material as Sustained Release Carrier"

_pharmaceutics, 2023, doi:10.3390/pharmaceutics15102434_

Round 1

Reviewer 1 Report

I have carefully reviewed the manuscript "Recent progress in diatom biosilica: a superior natural drug delivery agent" submitted to Pharmaceutics. I appreciate the authors' effort in addressing an important topic in the field of drug delivery systems and applications based on biosilica. However, after thorough evaluation, I cannot recommend this manuscript for publication until major revision.

Firstly, one notable issue is the presentation of figures in the manuscript. It is crucial for figures to be clear and legible to effectively convey the study findings. Therefore, I suggest that the authors revise Figures 2,4,5 to ensure readability and provide an updated version of the manuscript with the revised figures.

Furthermore, while the abstract provides an overview of the selected papers and highlights the heterogeneity in methodological approaches, it lacks a comprehensive analysis of the role of MS-DDS compared to other delivery systems. As a review article, it is expected to provide scientific insights and draw meaningful conclusions based on the available research data. This will enhance the significance and contribution of the manuscript to the field.

Additionally, the conclusion should be revised to provide a more definitive and focused summary of the key findings and implications of the reviewed studies. It is important for a review article to offer valuable insights and recommendations for future research or clinical applications. The authors should consider discussing the challenges and opportunities in translating MS-DDS into clinically applicable approaches and propose potential strategies for bridging the gap between experimental studies and commercialization.

Author Response

1. Firstly, one notable issue is the presentation of figures in the manuscript. It is crucial for figures to be clear and legible to effectively convey the study findings. Therefore, I suggest that the authors revise Figures 2,4,5 to ensure readability and provide an updated version of the manuscript with the revised figures.

> The authors agree with the reviewer's comments. We revised Figures 2, 4, and 5 to ensure readability. Please check page 4, line 144-148 for Figure 2; page 7, line 232-237 for Figure 4; and page 10, line 327-333 for Figure 5 (marked with yellow color). And, added references are marked in blue color (page 16, line 633 and 634).

2. Furthermore, while the abstract provides an overview of the selected papers and highlights the heterogeneity in methodological approaches, it lacks a comprehensive analysis of the role of MS-DDS compared to other delivery systems. As a review article, it is expected to provide scientific insights and draw meaningful conclusions based on the available research data. This will enhance the significance and contribution of the manuscript to the field.

> Thank you for reviewer's comments. Based on the reviewer's opinion, the abstract was revised. Please check page 1, line 21-26 marked with yellow color.

3. Additionally, the conclusion should be revised to provide a more definitive and focused summary of the key findings and implications of the reviewed studies. It is important for a review article to offer valuable insights and recommendations for future research or clinical applications. The authors should consider discussing the challenges and opportunities in translating MS-DDS into clinically applicable approaches and propose potential strategies for bridging the gap between experimental studies and commercialization.

> We sincerely appreciate the comments of our valued reviewers. The conclusion was revised according to the reviewer's comments. Revised parts are marked with yellow (page 13, line 478-492)

Reviewer 2 Report

The authors of the review article titled, 'Recent progress in diatom biosilica: a superior natural drug delivery agent' have done a great job in compiling information related to a biologically derived mesoporous silica and its application towards drug delivery. 

General Comment: A general statement or information about scenario would you use a diatom biosilica vs. traditional silicas? How would the biosilicas break down when administered parenterally? Considering traditional silicas do not solubilize, how would these be eliminated from the body? Add to the manuscript. From Line 387, it seems that there would be some amount of residual content of the biosilicas? Would these cause any long term damage to the cells or organs? 

Lines 185-188: I would suggest that the authors revisit the paper and rephrase the lines. Spectroscopic methods for formulation development?

Author Response

1. General Comment: A general statement or information about scenario would you use a diatom biosilica vs. traditional silicas? How would the biosilicas break down when administered parenterally? Considering traditional silicas do not solubilize, how would these be eliminated from the body? Add to the manuscript. From Line 387, it seems that there would be some amount of residual content of the biosilicas? Would these cause any long-term damage to the cells or organs?

> We sincerely appreciate the comments of our valued reviewers. Previous animal studies based on mice and rats have reported that biosilica injected into the body decomposes and some of it is excreted through the kidneys (Delalat et al., 2015; Borak et al., 2012). In addition, it was confirmed that it doesn’t cause damage to major organs (i.e., brain, heart, kidney, liver and lung) and tissues (Delalat et al., 2015; Borak et al., 2012; Zhong et al., 2021). Although research has not yet been conducted to determine the exact decomposition mechanism of biosilica, it has been reported that MSN, a synthetic silica, is decomposed in the cytoplasm and lysosomes of human umbilical vein endothelial cells (HUVEC) (Zhai et al., 2012). Therefore, it is assumed that biosilica injected into the body also can be decomposed in HUVEC, but additional research is needed to make an accurate identification. We have added the relevant content above to the discussion. Please check page 11, lines 392-408 marked with yellow color. And, added references are marked in blue color (page 19 and 20, line 803-820).

- Delalat, B.; Sheppard, V.C.; Rasi Ghaemi, S.; Rao, S.; Prestidge, C.A.; McPhee, G.; Rogers, M.-L.; Donoghue, J.F.; Pillay, V.; Johns, T.G.; Kröger, N.; Voelcker N.H. Targeted Drug Delivery Using Genetically Engineered Diatom Biosilica. Nat. Commun. 2015, 6, 8791. https://doi.org/10.1038/ncomms9791.

- Borak, B.; Biernat, P.; Prescha, A.; Baszczuk, A.; Pluta, J. In Vivo Study on the Biodistribution of Silica Particles in the Bodies of Rats. Adv. Clin. Exp. Med. 2012, 21, 13–18.

- Zhong, D.; Du, Z.; Zhou, M. Algae: A Natural Active Material for Biomedical Applications. View 2021, 2, 20200189. https://doi.org/10.1002/VIW.20200189.

- Zhai, W.; He, C.; Wu, L.; Zhou, Y.; Chen, H.; Chang, J.; Zhang, H. Degradation of Hollow Mesoporous Silica Nanoparticles in Human Umbilical Vein Endothelial Cells. J. Biomed. Mater. Res. Part B Appl. Biomater. 2012, 100, 1397–1403. https://doi.org/10.1002/jbm.b.32711.

2. Lines 185-188: I would suggest that the authors revisit the paper and rephrase the lines. Spectroscopic methods for formulation development?

> Thank you for reviewer's comments. The mentioned parts were revised to reflect reviewer's comments. Please check page 6, lines 187-192 marked with yellow color.

Reviewer 3 Report

The manuscript submitted by Lim et al. is a review article on biosilica as a delivery systems. The topic is suitable for Pharmaceutics. However, the manuscript may be accepted for publication after careful revision:

1.   The title does not match the content of the manuscript. In particular, the title states "...: a superior natural delivery agent". However, the article deals exclusively with the delivery of anticancer drugs and the relevance of cancer therapy (see the beginning of the introduction). In this regard, the title should be corrected in accordance with the content.

2. The authors admire biosilica-based delivery systems and present them only in a positive light without providing critical conclusions. In particular, the authors do not discuss at all the possible elimination routes of such systems from the body, their immunogenicity and properties related to blood clearance, e.g. by macrophage uptake, when used in injections.

3.  The ways how these systems can be improved and their place and strong sides among other delivery systems, for example biodegradable nanoparticles, should be discussed.

4.  Prospects and directions for further development and improvement for such materials should be highlighted in the conclusions.

5.  Table 1, last row. What does specific surface area "high" mean? Is it tens or hundreds of m2/g? Specific ranges should be given as indicated for particle sizes.

Author Response

1.   The title does not match the content of the manuscript. In particular, the title states "...: a superior natural delivery agent". However, the article deals exclusively with the delivery of anticancer drugs and the relevance of cancer therapy (see the beginning of the introduction). In this regard, the title should be corrected in accordance with the content.

> We sincerely appreciate the valuable reviewer's comments. In this review paper, we describe the potential of biosilica as a sustained-release agent to improve the bioavailability of various types of drugs. Therefore, based on the reviewer's opinion, the title of this review was revised to "Recent progress in diatom biosilica: a natural nanoporous silica material as sustained release carrier". Please check page 1, line 2-3 (marked with yellow color).

2. The authors admire biosilica-based delivery systems and present them only in a positive light without providing critical conclusions. In particular, the authors do not discuss at all the possible elimination routes of such systems from the body, their immunogenicity and properties related to blood clearance, e.g. by macrophage uptake, when used in injections.

> Thank you for reviewer's comments. Based on the reviewer's opinion, conclusion and discussion were revised on the comments. First, a paragraph related to the biodegradability and cytotoxicity of biosilica was added to the discussion (section 4.1; marked with yellow in page 11, line 392-408). And, at the end of the conclusion, the problems that must be solved for the commercialization of biosilica are described. Please check page 13, line 478-492 marked with yellow color.

3.  The ways how these systems can be improved and their place and strong sides among other delivery systems, for example biodegradable nanoparticles, should be discussed.

> Thank you for reviewer's comments. Based on the reviewer's comments, we have added a section in the discussion to compare the advantages between diatoms and other biodegradable DDS platforms. Please check pages 11 and 12, lines 409-420 (marked with yellow color).

4.  Prospects and directions for further development and improvement for such materials should be highlighted in the conclusions.

> Thanks for the reviewer's valuable comments, and the authors agree with the opinion. We have added to the conclusion a section about further developments and improvements needed for biosilica to become a useful DDS platform. Please check page 13, line 478-492 marked in yellow color.

5.  Table 1, last row. What does specific surface area "high" mean? Is it tens or hundreds of m2/g? Specific ranges should be given as indicated for particle sizes.

> We appreciated the reviewer’s valuable comments. The last row of Table 1 was revised to reflect the reviewer's opinion. Please check page 5 (marked with yellow color). And, added references are marked in blue color (page 17 and 18, line 700-709)

Round 2

Reviewer 3 Report

The authors responded to my questions and comments and performed a corresponding revision. In my opinion, the manuscript can be accepted for publication.